# Increasing Frequency of G275E Mutation in the Nicotinic Acetylcholine Receptor *α6* Subunit Conferring Spinetoram Resistance in Invading Populations of Western Flower Thrips in China

**DOI:** 10.3390/insects13040331

**Published:** 2022-03-28

**Authors:** Li-Na Sun, Xiu-Jing Shen, Li-Jun Cao, Jin-Cui Chen, Li-Jun Ma, San-An Wu, Ary Anthony Hoffmann, Shu-Jun Wei

**Affiliations:** 1Beijing Key Laboratory for Forest Pest Control, College of Forestry, Beijing Forestry University, Beijing 100083, China; sunlina19952005@163.com; 2Institute of Plant Protection, Beijing Academy of Agriculture and Forestry Sciences, Beijing 100097, China; shenxj1230@163.com (X.-J.S.); gmatjhpl@163.com (L.-J.C.); chenjincui1314@126.com (J.-C.C.); ma_lijun@foxmail.com (L.-J.M.); 3School of BioSciences, Bio21 Institute, University of Melbourne, Parkville, VIC 3052, Australia; ary@unimelb.edu.au; 4Department of Chemistry and Bioscience, Aalborg University, 9220 Aalborg, Denmark

**Keywords:** biological invasion, KASP, molecular diagnostics, pesticide resistance, spinosyns

## Abstract

**Simple Summary:**

The western flower thrips (WFT) *Frankliniella occidentalis* (Pergande) (Thysanoptera: Thripidae) is an important invasive pest in agriculture and forestry. It has developed resistance to a frequently used pesticide spinetoram world widely, including the invading area of China. However, the mechanism of resistance to spinetoram is unclear in China. In this study, we found the presence of the G275E mutation in the nicotinic acetylcholine receptor *Foα6* in the early invading populations, which has now increased to a high frequency in China. There was a correlation between the frequency of the G275E mutation and resistance to spinetoram as characterized by median lethal concentration. Our results showed that G275E mutation is one of the mechanisms conferring spinetoram resistance in invading populations in China, as in many other countries. Our study highlights the rapid spread of the G275E mutation in China in the 2009–2021 period.

**Abstract:**

The western flower thrips *Frankliniella occidentalis* (Pergande) (Thysanoptera: Thripidae) is an important invasive pest worldwide. Field-evolved resistance to the pesticide spinetoram is an increasing problem in the chemical control of this pest. Here, we examined changes in the frequency of a genetic mutation associated with spinetoram resistance, the G275E mutation in the acetylcholine receptor *Foα6*, in 62 field populations collected from 2009 to 2021 across areas of China invaded by this pest. We found a low frequency of the G275E mutation in populations collected at the early invasion stage, in contrast to a high frequency in native USA populations. However, the frequency of the G275E mutation has increased to a high level in recently collected populations, with the mutation becoming fixed in some populations. There was a correlation between the frequency of the G275E mutation and resistance to spinetoram as characterized by median lethal concentration, although two populations were outliers. These results showed that G275E mutation is one of the mechanisms conferring spinetoram resistance in many invading populations in China. Ongoing dispersal of the WFT may have facilitated a rapid increase in the G275E mutation across China. Our study highlights the rapid evolution of pesticide resistance in an invasive species and points to a useful marker for molecular diagnostics of spinetoram resistance.

## 1. Introduction

The western flower thrips (WFT), *Frankliniella occidentalis* (Pergande) (Thysanoptera: Thripidae), is a polyphagous and widespread invasive pest of agriculture and horticulture [1]. It can severely damage more than 500 host crops by direct feeding and oviposition as well as through transmitting plant viruses in both greenhouse and field situations [2]. The WFT originated from North America and has spread extensively through international trade since the late 1970s [3]. WFT has now successfully invaded many countries on all continents except Antarctica [3]. In China, it was first intercepted in Kunming, Yunnan province in 2000, although it was first reported as an invader in pepper cultivated in the Beijing area in 2003 [4]. Since then, WFT has rapidly colonized all suitable areas of China and become a common pest of vegetable crops and flowers [5].

Following rapid invasion of WFT, chemical control became the most reliable measure to control outbreaks of this species in the absence of natural predators and effective physical control measures. The insecticides used to control WFT since the 1960s include organochlorines, pyrethroids, carbamates, neonicotinoids, avermectins and spinosyns [2,6]. However, as WFT rapidly proliferated and control efforts increased, the inevitable problem of insecticide resistance has emerged. WFT populations from California greenhouses had probably already developed resistance to some classes of insecticides before becoming invasive [7]. Invasive populations may therefore have carried resistance alleles, and WFT in China initially showed a low level of susceptibility to some major insecticides [8]. Currently, some WFT populations have developed resistance to spinosad, spinetoram, abamectin, emamectin-benzoate, thiamethoxam and β-cypermethrin [9]. In particular, there have been a number of cases of rapid and strong resistance developing to spinetoram, with high median lethal concentration (LC_50_) values and a resistance ratio of >16,000 compared to a susceptible strain [10].

Spinetoram belongs to a group of insecticides known as spinosyns (IRAC class 5) and acts through activation of the nicotinic acetylcholine receptors (nAChRs) in the nervous system of insect pest species [11,12]. Spinosyns are extensively used to control a range of agricultural pests, such as thrips [13,14], *Helicoverpa armigera* (Hübner) (Lepidoptera: Heliothinae) [15] and *Plutella xylostella* (Linnaeus) (Lepidoptera: Plutellidae) [16]. Resistance mechanisms are mainly due to a single major gene mutation, which involves the subunit *α6* of nAChR in *Drosophila melanogaster* (Meigen) (Diptera: Drosophilidae) [17]. Glycine at amino acid position 275 of the *α6* subunit was mutated to glutamic acid in the resistant strains of both WFT [18,19,20] and the melon thrips, *Thrips palmi* (Karny) (Thysanoptera: Thripidae) [21]. Moreover, Hiruta et al. [22] found G275 mutated to valine in a spinosad-resistant strain of the flower thrips, *Frankliniella intonsa* (Trybom) (Thysanoptera: Thripidae). In Australia, a G275 mutation (G275E) has also been associated with field resistance to spinetoram in WFT [19]. However, three susceptible strains and two resistant strains of WFT from China and the United States showed no differences in the cDNA of *α6* [23], which indicates that this subunit is not the only mutation involved in spinosad resistance.

Here, we genotyped the G275E mutation of nAChR *α6* (*Foα6*) in sixty field populations from China and two populations from the United States collected during 2009–2021 by Kompetitive Allele-Specific PCR (KASP), which is a quick and efficient molecular diagnostic method [24,25]. We also compared our genotyping results with a conventional bioassay for spinetoram resistance in seven field populations to determine the contribution of the G275E mutation to spinetoram resistance. Our study highlights the rapid spread of the G275E mutation in China in the 2009–2021 period, with potential applications to ongoing monitoring of resistance to spinetoram in WFT populations.

## 2. Materials and Methods

### 2.1. Sample Collection and DNA Extraction

WFT collections were obtained between 2009 and 2021, coinciding with the period of spread of this pest in China. Adults were collected from 60 locations across their distribution in China with samples from an additional two locations from the US (collected in 2010) also included (Figure 1, Appendix A). Chinese samples of WFT were divided into three collection periods coinciding with invasion stages (Appendix A). Samples were obtained by collecting adults throughout an area of at least 1000 m^2^ and then taken to the laboratory in sealed bags. To reduce the likelihood of the collected thrips being close relatives, each individual was collected two meters from another individual. Female adults were used for population genetic analysis because of male-haploid from thrips’ parthenogenetic reproduction. All specimens were kept in absolute ethanol at −80 °C before DNA extraction.

Ten to eighty-seven female adults of WFT from each population were analyzed (Appendix A), with a total of 1508 individuals processed. Genomic DNA was extracted from individual WFTs using a high-throughput method; an individual was placed into a well of a 96-well plate containing 80 μL lysis buffer, 1 μL proteinase K [26] and a steel ball (2 mm in diameter). Then, specimens were ground up with a high-throughput tissue grinder (MiniG 1600, SPEX SamplePrep, Metuchen, NJ, USA) for 10 min at 1500 rpm. The grounded specimens were incubated at 65 °C for 30 min followed by 95 °C for 10 min. The supernatant extract containing genomic DNA was kept for subsequent usage after centrifugation for 1 min with a speed of 4000 rpm and stored at −20 °C.

### 2.2. Bioassay

A Spinetoram (Exalt 6 g·L^−1^ SC, recommended application rate in 1.5–3.0 μL·m^−2^, Dow AgroSciences Company, Midland, MI, USA) preparation was used in this study, with the chemical applied through a modified spray tower [27] to test the susceptibility of seven field populations of WFT (Appendix A). The recommended concentration of spinetoram for field spray is 10 mg/L. We serially diluted spinetoram into five concentrations ranging from 0.0625–50 mg/L with water containing 0.1% Triton X-100 (Beijing Solar BioScience and Technology Limited Company, China). Cucumber leaves which had never been exposed to pesticides were cut into a circle and pressed into plastic cups (6 cm in diameter and 4 cm in height) with a layer of 1% agar (0.2 cm in height) to seal the leaves and keep them fresh. About 25 female adults were then placed into a cup with an aspirator. The thrips in a cup were anesthetized with carbon dioxide and then cups were transferred to the spray tower. We applied 2 mL of an insecticide solution (or control) to the thrips in the cup. Then, the cups were quickly covered with a tissue and a lid with a circular hole (1.5 cm in diameter). Three replicate cups were tested per concentration and the 0.1% Triton X-100 control treatment. The treated thrips were kept under 25 °C, 60–70% relative humidity and a photoperiod of 16 h:8 h (L:D). Mortality was recorded 48 h after treatment. Thrips were scored as dead when touched with a brush tip and unable to move.

### 2.3. KASP Primer Design

Two allele-specific forward primers and one common primer were designed by LGC Genomics (UK) based on the DNA segment of the nAChR *α6* subunit gene around the G275E mutation [20]. Each forward primer has one of fluorophore (FAM: GAAGGTGACCAAGTTCATGCT, HEX: GAAGGTCGGAGTCAACGGATT) attached as a tail and the sequences are as follows: susceptible allele forward primer, 5′-GAACATGATGCAGTTGAAGTAAGTTC-3′; resistant allele forward primer, 5′-ATGAACATGATGCAGTTGAAGTAAGTTT-3′; and common primer, CGGCCACTGCCTGCGTCTGTTT.

### 2.4. KASP Assay of G275E Mutation

The DNA samples were assayed in a 384-well plate. The total volume of KASP reaction mixtures was 3 µL, containing 1.5 µL 2 × KASP master mixture (Standard Rox, LGC Genomics, Hoddesdon, Herts., UK), 12 µM of each allele-specific forward primer, 30 µM of common primer, at least 1 ng genomic DNA and 1.5 µL ddH_2_O.

The KASP reaction was performed on a Hydrocycler-16 machine (LGC Genomics, Hoddesdon, Herts., UK) with the following program: 15 min at 94 °C, followed by 10 touchdown cycles (20 s at 94 °C; 25 s at touchdown at 68 °C initially and decreasing by 0.6 °C per cycle) and then followed by 26 cycles of amplification (20 s at 94 °C; 60 s at 62 °C). Then, the fluorescence signal was read on a Pherastar scanning machine (BMG Latech, Offenburg, Germany). We used SNP viewer 2 (LGC Genomics, UK) to check the KASP assay and output the results.

### 2.5. Data Analysis

For the bioassay data, the LC_50_ (median lethal concentration) was computed for each WFT population tested. Parameters for the regression model and LC_50_ values were estimated in DPS software [28]. The resistance ratio (RR) was calculated by dividing the LC_50_ of a population by the lowest LC_50_ obtained from the WFT populations. Differences between populations were considered significant when the 95% confidence intervals (95% CIs) for the LC_50_ values did not overlap.

Using the KASP results, tests for deviation from the Hardy-Weinberg equilibrium (HWE) at the mutation G275E were performed based on the chi-square statistic with the R package *genetics* version 1.3.8 after applying a Bonferroni correction (Baltzegar et al., 2021). A one-way analysis of variance (ANOVA) was performed to test for differences in the frequencies of the resistance allele in the three invasion stages using IBM SPSS Statistics version 20 (IBM, Armonk, NY, USA), with pairwise comparisons undertaken based on Tukey’s honestly significant difference (HSD) test. A linear regression was undertaken to investigate whether the frequency of the G275E mutation could be predicted by the (log transformed) LC_50_ of that population using GraphPad Prism version 9.0 (GraphPad Software, San Diego, CA, USA). We also considered these associations based on the frequency of the resistant homozygote and summed resistant and susceptible homozygotes (to reflect possible different dominance relationships between the resistant allele over the susceptible allele).

## 3. Results

### 3.1. Resistance of Seven Field Populations of WFT to Spinetoram

We performed bioassays on seven field populations. The LC_50_ values of these populations ranged from 0.24 mg·L^−1^ (BJDJ, 2017) to 52.9 mg·L^−1^ (BJDJ, 2020) (Table 1, Figure 1). The LC_50_ value of BJDJ was significantly lower than that of other populations. There were two other populations (BJPG and BJDS) with LC_50_ values lower than 1.0 mg·L^−1^. The BJHD population had a higher level of resistance (LC_50_ = 8.7 mg·L^−1^, RR = 36.25), while the BJDQ (LC_50_ = 42.32 mg·L^−1^, RR = 176.33) and BJDJ (LC_50_ = 52.90 mg·L^−1^, RR = 220.42) populations had very high levels of resistance.

### 3.2. Frequency of G275E Mutations in Spatial and Temporal Populations

We genotyped 1508 individuals for the G275E mutation by KASP. The only genotype detected in the USDZ population was the homozygous resistant genotype (RR). Genotype frequencies in the USA population (USJZ) were 64.7% (RR), 17.6% (RS-heterozygote) and 17.6% (SS-susceptible homozygote). In the 2009–2013 samples from China, the highest (16.7%) frequency of the resistant homozygote (RR) was detected in the XJKL population. No RR or RS individuals were detected in five populations (BJYL, SDQD, BJDX, BJHD2011 and BJHD2013). Genotype frequencies in four other populations (XJKL, BJMT, YNKJ and YNHH) deviated from HWE (Figure 2a). In 2014–2018, the resistance homozygous (RR) frequency was highest (81.8%) in the BJDQ2017 population and also over 50% in the BJDQ2018 population. However, the RR genotype was still relatively rare in the other populations, and eight populations only had the sensitive SS homozygote. Genotype frequencies in three populations (BJCH, BJHD2017 and BJDQ2018) deviated from HWE (Figure 2b). By 2020–2021, the frequency of the RR genotype was over 50% in more populations than in the first period (2009–2013) and second period (2014–2018). The SCCD population only had RR homozygotes, while three populations only had SS homozygote individuals. The genotype frequencies in three populations (BJPG, SCDY and BJYZ2021) deviated from HWE (Figure 2c).

The ANOVA results showed that the frequencies of resistance allele were significantly different across periods (F_2,57_ = 7.138, *p* = 0.011, Figure 3). There was a significant difference in the resistance allele frequencies between population groups collected from 2014–2018 and from 2020–2021 (*p* = 0.012, 95% CI = 4.6045–45.1037, Figure 3). The frequencies of resistance allele increased, being significantly higher in the third period (2020–2021, 37.14%) than in the two earlier periods (8.89% in 2009–2013 and 12.29% in 2014–2018).

### 3.3. Association between G275E Mutation and Resistance of WFT to Spinetoram

When considering all seven populations, the linear regression analysis showed no strong association between (log transformed) resistance to spinetoram and the frequency of the resistance allele (R^2^ = 0.3099, *p* = 0.1943, Figure 4a), the frequency of the RR genotype (R^2^ = 0.2200, *p* = 0.2883, Figure 4b) or the summed frequency of the RR and RS genotypes (R^2^ = 0.3315, *p* = 0.1761, Figure 4c). When we removed the data of BJDJ2020 and BJHD2021 populations, the linear regression analysis did show an association between resistance and the frequency of the resistance allele (R^2^ = 0.9543, *p* = 0.0042, Figure 4d), the frequency of the RR genotype (R^2^ = 0.8966, *p* = 0.0146, Figure 4e) and the summed frequency of the RR and RS genotypes (R^2^ = 0.8799, *p* = 0.0183, Figure 4f).

## 4. Discussion

In this study, we considered the detection and distribution of genotypes carrying the nAChR *α6* G275E mutation in field populations of WFT during 2009–2021. The results will first be discussed in terms of the distribution of resistance genotypes at different invasion stages and following the local registration of spinetoram against WFT. We then discuss the contribution of the G275E mutation to possible mechanisms underlying spinetoram resistance. Finally, we consider the new high-throughput KASP diagnostic assay developed here as an effective way of detecting the G275E mutation. This diagnostic assay can be used to monitor resistance alleles in field populations in support of integrated pest management (IPM) programs that aim to manage resistance.

### 4.1. The G275E Resistance Mutation of WFT to Spinetoram in China

Low frequencies of RR and RS genotype of the nAChR *α6* G275E mutation were detected in all field WFT populations from China. The G275E mutation was at a low level in the first period (2009–2013) and also in the second period (2014–2018) except in one population in Daxing, Beijing. However, the frequency of the resistant allele markedly increased in the third period by a factor of 4 over the first period. Bioassay results show the same pattern when LC_50_ values are compared across studies; the susceptibility of populations to spinetoram was high before 2014 with an average LC_50_ value below 0.07 mg·L^−1^ [29]. From 2015 to 2018, the susceptibility of WFT to spinetoram decreased in some populations, with an LC_50_ value for northern populations higher than this value, especially for those from Beijing and the Shandong province [8,9,10]. In our study, the LC_50_ values of populations in the Beijing area were in the range 0.65–52.9 mg·L^−1^ in 2020–2021, a further change that points to susceptibility decreasing rapidly after the second period (2014–2018).

### 4.2. Spinetoram Resistance Mechanisms in WFT

In the association analysis linking the LC_50_ values of seven populations in the Beijing area to the frequency of the G275E mutation, we only found a weak association unless we removed data from two populations. We suspect that mutation G275E of *Foα6* represents an important target-site mutation leading to spinetoram resistance as also noted for the tomato leafminer, *Tuta absoluta* [30], and consistent with other studies on WFT. The presence of two populations that do not fit this pattern points to another important mechanism being involved in spinetoram resistance in WFT, such as the expression of truncated nAChR *α6* variants [31]. In previous studies, metabolism-based resistance mechanisms to spinosyns have been ruled out in laboratory strains of WFT [18,32,33], but we suspect that this requires further study. Multiple mechanisms are known to be involved in other pests such as *T. palmi*, where spinosyn resistance in the field is conferred by both reduced sensitivity of TP*α6* and by cytochrome P450-mediated detoxification [21,34]. However, the detailed genetic basis of spinosyn resistance in these cases has not yet been established, and it is intriguing that the outlier populations have been identified from one region where other populations seem to fit the general pattern of increasing resistance with an increasing frequency of the mutation.

### 4.3. Association of the G275E Mutation with WFT Invasion and Spread

We detected a high frequency of the resistant mutation in early period collections of American WFT populations, in contrast to a low mutation frequency in the early Chinese WFT populations. With the spread of the WFT in China, the resistance of WFT increased, along with the frequency of the resistant mutation. Ongoing WFT dispersal across the time periods may have accelerated the development of resistance. We have previously noted a population structure in WFT consistent with anthropogenic transmission which can span great distances, introducing new alleles that can then further increase under selection [5]. We have also previously noted the spread of resistance in China in other pests including *T. palmi* and spider mites [35,36].

### 4.4. A KASP Diagnostic Assay for the G275E Mutation

For low cost and high throughput genotyping of G275E mutation in WFT, we designed a KASP assay that can genotype the G275E mutation in individual insects from field-collected populations [37]. The diagnostic assay differed in sequence only at the mutation site, with one primer complementary to the susceptible wild-type allele and the other primer complementary to the mutant-resistant allele. This assay allowed us to rapidly genotype 62 field populations from past collections to reveal that 44 out of 62 field populations harbored the G275E mutation. The KASP assay can also be used for high-throughput screening of resistance alleles from small samples. Incorporating our molecular diagnostic assay as an adjunct in support of IPM-based resistance management will provide timely and precise information on resistance frequencies and aid successful management [24,38]. This is particularly important given the high level of variability in resistant allele frequency across populations, although we acknowledge that the likely presence of multiple resistance mechanisms means that a low incidence of the R allele does not preclude resistance being present in a population.

## 5. Conclusions

We examined spatial and historical variation of G275E mutation in the invasion populations of the WFT across China. Association analysis showed that G275E mutation is one of the mechanisms conferring spinetoram resistance in invading populations in China, although other mutations can also be important. We found that the G275E mutation present in early invading populations of the WFT increased in recent years. Ongoing dispersal of the WFT may have facilitated a rapid increase in the G275E mutation across China. Our study emphasizes the importance of molecular diagnostics of pesticide resistance in the invading populations to manage biological invasions.

## Figures and Tables

**Figure 1 insects-13-00331-f001:**
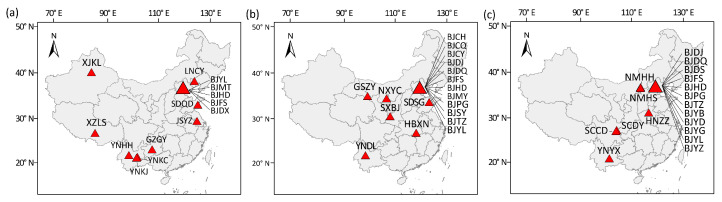
Sample locations (red triangle) of *Frankliniella occidentalis* populations collected in (**a**) 2009–2013, (**b**) 2014–2018 and (**c**) 2020–2021. The larger red triangle represents multiple populations collected from Beijing.

**Figure 2 insects-13-00331-f002:**
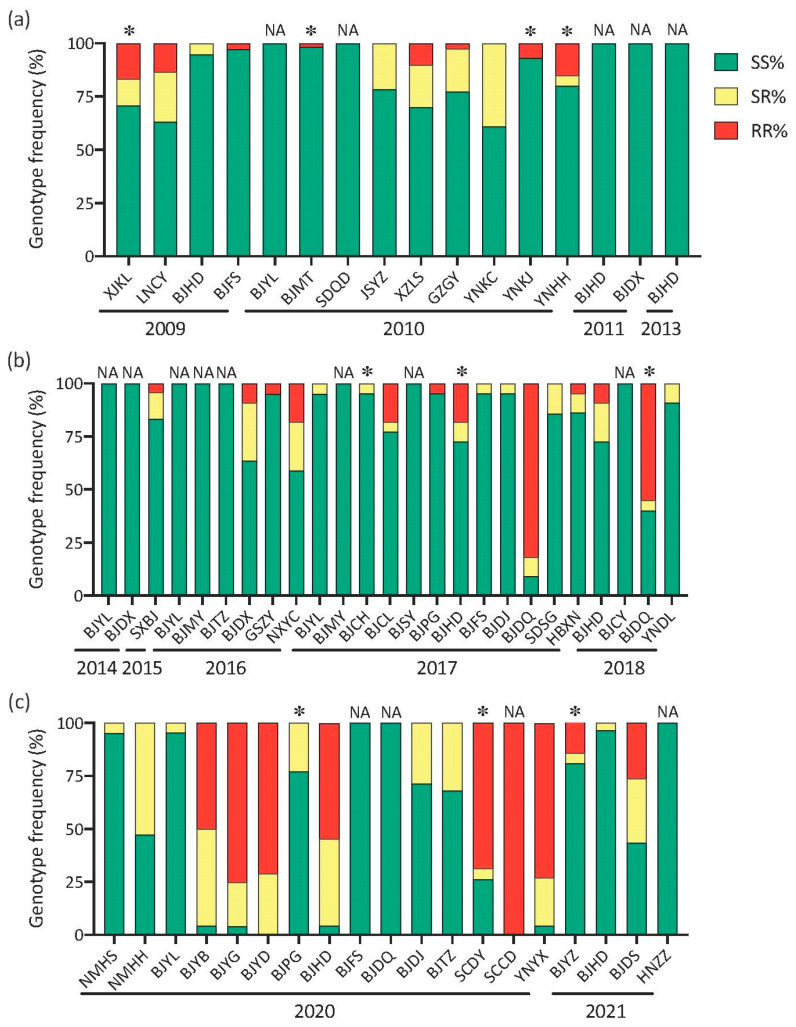
Genotype frequencies of G275E mutation in *Frankliniella occidentalis* populations during (**a**) 2009–2013, (**b**) 2014–2018 and (**c**) 2020–2021. RR, resistant homozygote genotype; RS, heterozygote genotype; SS, sensitive homozygote genotype. The “*” represents deviations from Hardy-Weinberg equilibrium (*p* < 0.05); NA represents a population consisting only of homozygous genotypes.

**Figure 3 insects-13-00331-f003:**
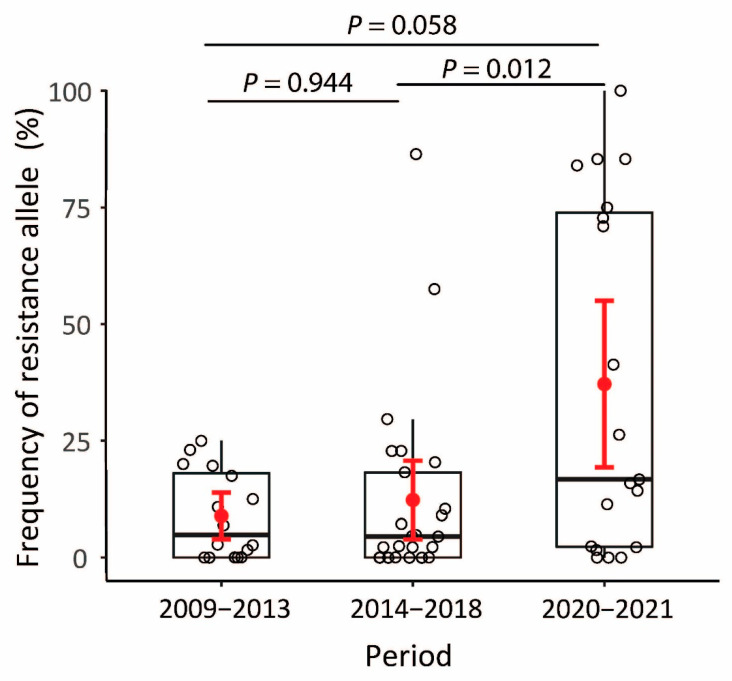
Plot of G275E-resistant allele frequencies in three collecting periods. The horizontal black solid line represents the median for each period; the red dot represents the mean. Red error bars indicate the standard error.

**Figure 4 insects-13-00331-f004:**
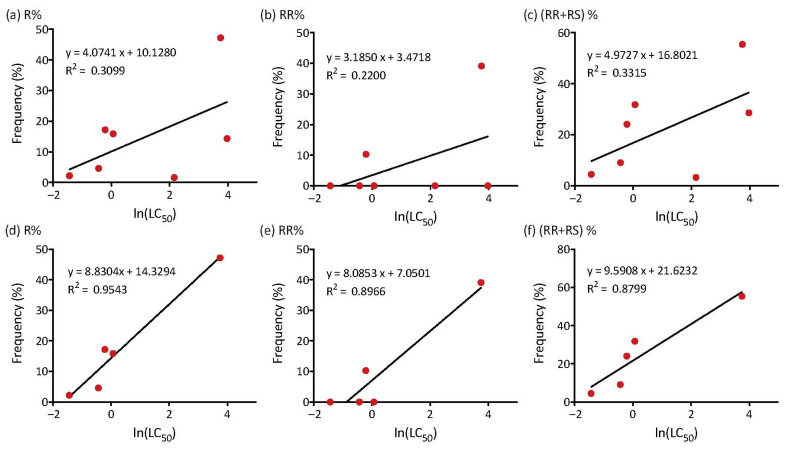
Regression between (ln) LC_50_ and the frequencies of (**a**) the resistant allele, (**b**) RR resistant genotype and (**c**) the sum of homozygous and heterozygote genotypes with the resistant G275E mutation across all populations tested. We have also provided the regressions when the BJDJ2020 and BJHD2021 populations are excluded (**d**–**f**).

**Table 1 insects-13-00331-t001:** Regressions for spinetoram responses and frequencies of G275E R/S genotypes in seven *Frankliniella occidentalis* populations.

Population	Year	Toxicity Regression Equation (*y* = b*x* + a)	LC_50_ (95% CI) (mg·L^−1^) *	SE (of b)	*χ* ^2^	*p*-Value	RR	R%	RR%	RS%
BJDJ	2017	2.76x + 6.73	0.24 (0.18–0.30) D	0.4399	0.5755	0.9020	1	2.2	0	4.5
BJDQ	2020	2.07x + 4.28	42.32 (35.82–50.19) A	0.1941	0.4737	0.9246	176.33	47.2	39.1	16.3
BJPG	2020	2.45x + 5.23	0.81 (0.62–1.04) BC	0.3847	1.4003	0.4965	3.38	17.2	10.3	13.8
BJDJ	2020	2.45x + 0.78	52.90 (43.87–64.28) A	0.2954	9.4655	0.0237	220.42	14.3	0	28.6
BJTZ	2020	3.35x + 4.90	1.07 (0.92–1.30) B	0.3627	0.4164	0.8120	4.46	15.9	0	31.8
BJDS	2021	2.13x + 5.40	0.65 (0.47–0.84) C	0.2250	9.4401	0.0510	2.71	4.6	0	9.1
BJHD	2021	0.91x + 4.15	8.70 (3.91–54.323) A	0.1078	19.1711	0.0018	36.25	1.6	0	3.3

SE, standard error; RR, resistance ratio. * Differences in LC_50_ were considered significant when the 95% CIs did not overlap, and these are marked by different letters.

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
