# Peer review of "Increasing Frequency of G275E Mutation in the Nicotinic Acetylcholine Receptor α6 Subunit Conferring Spinetoram Resistance in Invading Populations of Western Flower Thrips in China"

_insects, 2022, doi:10.3390/insects13040331_

Round 1

Reviewer 1 Report

These are my main comments on the manuscript (Insects-1636733) entitled “Increasing frequency of G275E mutation in the nicotinic acetyl-2 choline receptor α6 subunit conferring spinetoram resistance in 3 invading populations of western flower thrips in China”. The manuscript examines we examined changes in the frequency of a genetic mutation associated with spinetoram resistance. The results showed a mutation associated with a major mechanism conferring spinetoram resistance in many invading populations in China. Following moderated revisions should be incorporated in the manuscript prior to acceptance.
L.23: Define LC50s
L.43: Keywords should be in alphabetic order. Also, keywords serve to widen the opportunity to be retrieved from a database. To put words that already are into title and abstracts makes KW not useful. Please choose terms that are neither in the title nor in abstract.
Ls.33-34: For each scientific name species, provide the order and family taxa. Correct in all manuscript.
L.50: The WFT originated…
L.59: … neonicotinoids, avermectins, and spinosyns
Ls.73, 75, 77, etc.: For these species provide the ID author names, order, and family taxa. Check in all manuscript.
L.103: … absolute ethanol at -80°C…
L.104: Change “analysed” by “analyzed”
L.107: Change “μl” by “μL”. Correct in all manuscript.
L.128: narcotized should be anesthetized
L.129: Change “ml” by “mL” 
Table 1: Provide the p-value by each probit analysis
Figure 3: Treatment should be grouped using letters according to Tukey’s HSD test. 
L.235: … in terms of the distribution…
Ls.260 and 265: Tuta absoluta and T. palmi should be in italic.
L.276: Delete “significantly”
L.298: Delete “In this study,”

Author Response

Comments and Suggestions for Authors

These are my main comments on the manuscript (Insects-1636733) entitled “Increasing frequency of G275E mutation in the nicotinic acetyl-2 choline receptor α6 subunit conferring spinetoram resistance in 3 invading populations of western flower thrips in China”. The manuscript examines we examined changes in the frequency of a genetic mutation associated with spinetoram resistance. The results showed a mutation associated with a major mechanism conferring spinetoram resistance in many invading populations in China. Following moderated revisions should be incorporated in the manuscript prior to acceptance.

L.23: Define LC50s

>> Response: Added.

L.43: Keywords should be in alphabetic order. Also, keywords serve to widen the opportunity to be retrieved from a database. To put words that already are into title and abstracts makes KW not useful. Please choose terms that are neither in the title nor in abstract.

>> Response: We have changed the keywords in lines 43-44: Biological invasion; KASP; molecular diagnostics; pesticide resistance; spinosyns

Ls.33-34: For each scientific name species, provide the order and family taxa. Correct in all manuscript.

>> Response: We had checked all species in this article and provided the order and family taxa, please see the line 47 – 48 and line74 – 82.

L.50: The WFT originated…

>> Response: The problem had be revised in line 50-51.

L.59: … neonicotinoids, avermectins, and spinosyns.

>> Response: In article line 59 we have changed “abamectin” to “avermectins”.

Ls.73, 75, 77, etc.: For these species provide the ID author names, order, and family taxa. Check in all manuscript.

>> Response: We had checked all species in this article and provided the ID author names order and family taxa, please see the line 74 – 82.

L.103: … absolute ethanol at -80°C…

>> Response: We have changed “in” to “at” in line 106.

L.104: Change “analysed” by “analyzed”.

>> Response: We have revised in line 108 of the original text.

L.107: Change “μl” by “μL”. Correct in all manuscript.

>> Response: We've checked this paper and changed all “μl”by “μL”.

L.128: narcotized should be anesthetized.

>> Response: We have changed “narcotized” to “anesthetized” in line 131.

L.129: Change “ml” by “mL”.

>> Response: We've checked this paper and changed all “ml” by “mL”.

Table 1: Provide the p-value by each probit analysis.

>> Response: Added.

Figure 3: Treatment should be grouped using letters according to Tukey’s HSD test.

>> Response: Revised.

L.235: … in terms of the distribution…

>> Response: We have revised in line 238 of the original text.

Ls.260 and 265: Tuta absoluta and T. palmi should be in italic.

>> Response: We have revised in line 263 and 264 of the original text.

L.276: Delete “significantly”.

>> Response: The “significantly” has be delete in the 279.

L.298: Delete “In this study,”.

>> Response: “In this study” had been deleted in line 301.

Reviewer 2 Report

The authors conducted a large-scale genotyping of G275E mutation in the nAChR a6 subunit (known as the target gene of spinosyn insecticides) for populations of western flower thrips (WFT) in China and USA using a KASP method (one of PCR-based genotyping methods) developed by them.

The authors compared genotype frequencies of G275E mutation in WFT populations of three different periods and claimed that frequency of resistant genotypes (RR and RS) highly increased in 2020-2021 period in China.  The authors also conducted a bioassay using a spinetoram (one of spinosyn insecticides) for seven WFT populations and evaluated the correlation between the frequency of resistant genotypes and spinetoram resistance level (LC50). As a result, the authors claimed that although no strong correlation was observed in the seven populations, correlation was observed by removing two outliers. Finally, based on the results, the authors claimed that the G275E mutation is associated with a major mechanism of spinosyn resistance in many populations in China while other mutation can also be associated.

The large-scale G275E genotyping data using the KASP method and the bioassay results are very interesting and would help for elucidating the mechanism of spinosyn resistance of WFT in field populations.
However, there are several issues that need to be resolved for publication.

Major issues

1. Regarding the G275E detection, the sensitivity and specificity of the KASP method relative to sequencing method (e.g. direct-sequencing) should be evaluated because the accuracy of the method is of critical importance in this study. For example, the evaluation by Badolo et al. (2012) (https://doi.org/10.1186/1475-2875-11-227) may be helpful.

2. The results of Table 1 and Figure 4 clearly indicate that it is unclear if the G275E mutation is a major mechanism of spinosyn resistance in WTF, because two highly resistant populations (BJDJ2020 and BJHD2021) are clear outliers while only one highly resistant population (BJDQ2020) fits the correlation. Regarding another spinosyn resistance mechanism of WTF, dominant expression of truncated nAChR a6 variants was previously reported by Wan et al. (2018) (https://doi.org/10.1016/j.ibmb.2018.05.002). They showed a clear correlation between expression ratio of truncated nAChR a6 variants and resistance ratio of spinosad (another spinosyn insecticide). So, the high resistance observed in BJDJ2020 and BJHD2021 may be due to the dominant expression of the truncated nAChR a6 variants (this may also be true for BJDQ2020). In summary, I think the results of this study indicate that the G275E mutation is one of mechanisms of spinosyn in WTF, however, it is unclear if the G275E mutation is a major mechanism (may contribute to only a weak resistance in WTF). So, the authors should modify descriptions at lines 22-25, 303-305 and add descriptions about the another mechanism reported by Wan et al. accordingly.

3. At lines 164-167, the authors claimed that pairwise comparisons by Tukey's HSD were also conducted for evaluating significant differences of genotype frequencies between each pair of the three periods. However no results were described (only the result of one-way ANOVA is described at lines 205-206).

4. The lines 298-303 in 'Conclusions' are copy&paste of the lines 233-238 which looks strange in the context. The sentences should be rewritten.

Minor issues

1. In Figure. 3, no description for the red asterisk.
2. typos
   - Line21: the acetylcholine -> the nicotinic acetylcholine
   - Line196: BJDQ -> BJDQ2018
   - Fig. 2(a):   BJYQ -> BJYL

Author Response

Comments and Suggestions for Authors

The authors conducted a large-scale genotyping of G275E mutation in the nAChR a6 subunit (known as the target gene of spinosyn insecticides) for populations of western flower thrips (WFT) in China and USA using a KASP method (one of PCR-based genotyping methods) developed by them.

The authors compared genotype frequencies of G275E mutation in WFT populations of three different periods and claimed that frequency of resistant genotypes (RR and RS) highly increased in 2020-2021 period in China. The authors also conducted a bioassay using a spinetoram (one of spinosyn insecticides) for seven WFT populations and evaluated the correlation between the frequency of resistant genotypes and spinetoram resistance level (LC50). As a result, the authors claimed that although no strong correlation was observed in the seven populations, correlation was observed by removing two outliers. Finally, based on the results, the authors claimed that the G275E mutation is associated with a major mechanism of spinosyn resistance in many populations in China while other mutation can also be associated.

The large-scale G275E genotyping data using the KASP method and the bioassay results are very interesting and would help for elucidating the mechanism of spinosyn resistance of WFT in field populations.

However, there are several issues that need to be resolved for publication.

Major issues

  1. Regarding the G275E detection, the sensitivity and specificity of the KASP method relative to sequencing method (e.g. direct-sequencing) should be evaluated because the accuracy of the method is of critical importance in this study. For example, the evaluation by Badolo et al. (2012) (https://doi.org/10.1186/1475-2875-11-227) may be helpful.

>> Response: We compared the KASP assay with Sanger sequencing results using G275E mutation of the Thrips palmi. The results of these two methods are identical. That is the first application of KASP in resistant mutation detection. The manuscript is under review in the Pest Management Science.

  1. The results of Table 1 and Figure 4 clearly indicate that it is unclear if the G275E mutation is a major mechanism of spinosyn resistance in WTF, because two highly resistant populations (BJDJ2020 and BJHD2021) are clear outliers while only one highly resistant population (BJDQ2020) fits the correlation. Regarding another spinosyn resistance mechanism of WTF, dominant expression of truncated nAChR a6 variants was previously reported by Wan et al. (2018) (https://doi.org/10.1016/j.ibmb.2018.05.002). They showed a clear correlation between expression ratio of truncated nAChR a6 variants and resistance ratio of spinosad (another spinosyn insecticide). So, the high resistance observed in BJDJ2020 and BJHD2021 may be due to the dominant expression of the truncated nAChR a6 variants (this may also be true for BJDQ2020). In summary, I think the results of this study indicate that the G275E mutation is one of mechanisms of spinosyn in WTF, however, it is unclear if the G275E mutation is a major mechanism (may contribute to only a weak resistance in WTF). So, the authors should modify descriptions at lines 22-25, 303-305 and add descriptions about the another mechanism reported by Wan et al. accordingly.

>> Response: Thanks. We agree with your opinion and revised related contents.

  1. At lines 164-167, the authors claimed that pairwise comparisons by Tukey's HSD were also conducted for evaluating significant differences of genotype frequencies between each pair of the three periods. However no results were described (only the result of one-way ANOVA is described at lines 205-206).

>> Response: Added, also see Figure 3.

  • The lines 298-303 in 'Conclusions' are copy&paste of the lines 233-238 which looks strange in the context. The sentences should be rewritten.

>> Response: This paragraph was rewritten.

Minor issues

In Figure. 3, no description for the red asterisk.

>> Response: Added.

typos

 - Line21: the acetylcholine -> the nicotinic acetylcholine

>> Response: Added.

 - Line196: BJDQ -> BJDQ2018

>> Response: Revised.

 - Fig. 2(a): BJYQ -> BJYL

>> Response: Revised.

Round 2

Reviewer 2 Report

The issues have been addressed in the revised manuscript.